# The effects of upper body blood flow restriction training on muscles located proximal to the applied occlusive pressure: A systematic review with meta-analysis

Kyriakos Pavlou[1]⊙, Vasileios Korakakis[2]⊙, Rod Whiteley[3]*, Christos Karagiannis[1], George Ploutarchou[1], Christos Savva[1]

1 Department of Health Science, European University Cyprus, Engomi, Nicosia, Cyprus, 2 Department of Population Health Sciences, School of Life Course & Population Sciences, Faculty of Life Sciences and Medicine, King's College London, London, United Kingdom, 3 Aspetar, Orthopaedic and Sport Medicine Hospital, Doha, Qatar

⊙ These authors contributed equally to this work.
* Rodney.Whiteley@aspetar.com

**Data Availability Statement:** All relevant data are within the paper and its Supporting information files.

## Abstract

### Background

Blood flow restriction combined with low load resistance training (LL-BFRT) is associated with increases in upper limb muscle strength and size. The effect of LL-BFRT on upper limb muscles located proximal to the BFR cuff application is unclear.

### Objective

The aim of this systematic review was to evaluate the effect of LL-BFRT compared to low load, or high load resistance training (LL-RT, HL-RT) on musculature located proximal to cuff placement.

### Methods

Six electronic databases were searched for randomized controlled trials (RCTs). Two reviewers independently evaluated the risk of bias using the PEDro scale. We performed a meta-analysis using a random effects model, or calculated mean differences (fixed-effect) where appropriate. We judged the certainty of evidence using the GRADE approach.

### Results

The systematic literature searched yielded 346 articles, of which 9 studies were eligible. The evidence for all outcomes was of very low to low certainty. Across all comparisons, a significant increase in bench press and shoulder flexion strength was found in favor of LL-BFRT compared to LL-RT, and in shoulder lean mass and pectoralis major thickness in favor of the LL-BFRT compared to LL-RT and HL-RT, respectively. No significant differences were found between LL-BFRT and HL-RT in muscle strength.

**Funding:** If accepted for publication, this research may have open access fees paid for by the Qatar National Library Open Access Research Fund. Aside from this, this research did not receive any specific grant from funding agencies in the public, commercial, or not-for-profit sectors. Qatar National Library Open Access Research Fund had no role in study design, data collection and analysis, decision to publish, or preparation of the manuscript. To be clear, the authors received no specific funding for this work.

**Competing interests:** The authors have declared that no competing interests exist.

## Conclusion

With low certainty LL-BFRT appears to be equally effective to HL-RT for improving muscle strength in upper body muscles located proximal to the BFR stimulus in healthy adults. Furthermore, LL-BFRT may induce muscle size increase, but these adaptations are not superior to LL-RT or HL-RT.

## Introduction

Muscular hypertrophy and muscle strength improvements have been traditionally linked to a heavy-load resistance training (HL-RT) program [1]. According to the American College of Exercise Medicine resistance training load of ~60–70% of the one repetition maximum (1RM) is required to achieve improvements in muscle strength and 70–85% of 1 RM for gains in muscle hypertrophy [1]. However, such high loads are frequently not attainable in the clinical setting due the characteristics associated with musculoskeletal conditions such as the healing process, pain, muscle weakness, and functional or loading limitations [2].

Recently, significant attention has been drawn to low load resistance training (LL-RT) combined with blood flow restriction (BFR), which involves a parallel partial restriction of the arterial flow and complete occlusion of the venous return of the exercised limb [3, 4]. BFR is applied by using inflatable cuffs with an individually adjusted amount of compressed air placed at the most proximal part of the exercised limb [3]. Usually, the BFR cuff is applied at the deltoid tuberosity for an upper limb application and at the gluteal fold for a lower limb application [4].

Mounting evidence suggests that the use of BFR combined with LL-RT (20–40% of 1RM) may offer an applicable alternative to exercise with heavy-loads in improvement of muscle size and muscle strength [5–8]. Interestingly, studies have shown that these adaptations may occur even after a period of only three weeks of application with a training frequency of 2–3 times a week [9, 10]. In addition, low-load BFR training (LL-BFRT) has been found to be equally effective to traditional strength training in patients with knee osteoarthritis [11, 12], after a knee surgery including an ACL reconstruction [13], and in patients with anterior knee pain [14].

The effectiveness of this training method in improving muscle strength and hypertrophy has been consistently reported in the literature, but the exact mechanisms of action are still under investigation [4, 15]. Several hypotheses have been proposed with the higher levels of metabolic stress due to ischemic/hypoxic conditions being the most plausible mediator through several physiological pathways [16]. These mechanisms may include increases in hormonal concentrations, increases within the components of the intracellular signaling pathways for muscle protein synthesis—such as the mTOR pathway, increases within biomarkers denoting satellite cell activity, and patterns in fiber type recruitment [17].

While a significant number of studies focusing on the application of BFRT in the lower extremity have been published, research in the upper extremity is sparse [18–21]. A plausible explanation can be attributed to the anatomical location of the large muscle groups of the upper extremity (e.g., pectoralis major, deltoid, latissimus dorsi) that humpers the proximal application of the cuff and the restriction of the blood flow in contrast to lower extremity where the large muscle groups are located mainly distal (e.g., quadriceps) to the applied occlusive pressure [15].

Recently emerging evidence in healthy individuals revealed promising outcomes indicating that the use of LL-BFRT may result in increased muscle strength and hypertrophy in muscles located proximal to the BFR cuff application [15–22]. However, the published research until

today remains sparse. Hence, the main objective of this systematic review of randomized controlled trials (RCTs) was to evaluate in healthy individuals the effectiveness of LL-BFRT in inducing muscle adaptations, such as changes in muscle size and strength, in muscle groups surrounding the shoulder girdle and located proximally to the BFR cuff and the applied occlusive pressure.

## Materials and methods

### Protocol and guidelines

This systematic review adhered to Preferred Reporting Items for Systematic Reviews and Meta-Analyses (PRISMA) [23] and followed the recommendations of the Cochrane Handbook for Systematic Reviews [24].

### Eligibility criteria

The primary eligibility criteria were formulated based on the Population, Intervention, Comparison, Outcome, Study design (PICOS) framework [25] and were predefined as follows:

1. Population: healthy individuals (no age restriction).

2. Intervention: performance of LL-BFRT of the upper limb with the BFR cuff or elastic band applied before exercise, the limb remained restricted until exercise completion, and employed a training protocol consisted of at least five sessions to allow sufficient time for measurable muscle adaptations [9].

3. Comparison: a comparator group performing low, medium, or high load exercise of the upper limb.

4. Outcomes: pre- and post-training measures of muscle size and/or strength of muscles located proximally to the shoulder.

5. Study design: randomized controlled trials or quasi-experimental study designs written in the English language.

### Information sources and search strategy

The search was conducted independently by two reviewers (KP and CK) from database inception to May 2022 using the following databases and clinical trial registries: MEDLINE (PubMed), CINAHL (EBSCO), SPORTDiscus (EBSCO), SCOPUS, EU clinical trials, and ClinicalTrials.gov. The search strategy consisted of MeSH terms and keywords (synonyms and abbreviations) related to the BFR and the shoulder or the upper limb, combined with MeSH terms for RCTs. The full search strategy is presented in S1 Table of S1 File. The reference lists, citation tracking results, and systematic reviews were manually searched to identify studies that were not found through database searching.

### Study selection and data extraction

Final search results were imported into EndNote and duplicates were removed. Two reviewers (KP and VK) independently evaluated titles and abstracts, and the full text of the potentially eligible studies was obtained and evaluated, while disagreements were resolved by a third reviewer (CS).

One author (KP) abstracted relevant details about study design, sample size, demographic characteristics, attrition rate, BFR and exercise protocol (type, frequency, occlusion

characteristics, training load, and duration), pre- and post-intervention means and standard deviation for any strength or muscle size measures, and main within- and between-group results (strength and muscle size). A second investigator (VK) reviewed all data for accuracy. In case of missing data authors were contacted via email (twice). Data presented only in graphs were converted and obtained by using WebPlotDigitizer software (Version 4.5, https://automeris.io/WebPlotDigitizer/).

## Quality assessment and risk of bias

Two investigators (KP and VK) independently evaluated study quality using the Physiotherapy Evidence Database (PEDro) scale which is considered as a valid and reliable tool for assessing the internal and external validity of RCTs [26, 27]. Discrepancies were resolved through consensus.

The PEDro scale consists of ten items assessing the randomization and allocation process, the blinding, the baseline comparability, and the study reporting [27]. Studies scoring $\geq 7/10$ were rated as "high quality", studies with a score 4 to 6/10 as "moderate quality", and those with score $\leq 3$ as "low quality" studies [12, 19]. A PEDro quality score $<7$ indicated a study as having a "high" risk of bias [28].

## Data analysis, synthesis, and intervention effect

Using the available outcome measures for muscle size and strength, we calculated standardized mean differences (SMDs) and the associated 95% confidence intervals (95%CI) where data from more than one study was available, after appraising the variability of clinical settings and methods used for strength and muscle size assessment. We calculated and presented mean differences (MDs) in the case of a single study data availability [29]. When two or more studies were available pair-wise meta-analyses were conducted and forest plots were presented if aggregate pooled estimates met the sample and intervention homogeneity criteria. We pooled pair-wise meta-analyses assessing the same muscle group, using similar methods of strength and muscle volume assessment, which evaluated muscle loading of comparable magnitude where recruited participants displayed comparable demographic characteristics, and after leave-one-out sensitivity analyses.

When only one study was available for an outcome an effect size (MD, fixed effect model) was calculated and presented. Results per outcome and muscle group were presented as summary tables. For the effect estimates the Review Manager V.5.3 statistical software of the Nordic Cochrane Collaboration was used and assuming methodological and setting heterogeneity between studies, a random effects meta synthesis was employed where applicable. Strength values were transformed in kilograms for analyses, where applicable.

If considerable between-group statistical heterogeneity was detected (i.e., $I^2 > 75\%$), we did not perform a meta-analysis [29], but evaluated the heterogeneity with sensitivity analyses by excluding studies with unexpectedly large treatment effect sizes and 'leave-one-out' exclusion, and studies presenting significant heterogeneity at baseline for participant characteristics. Given the small number of included studies, assessment of reporting bias with a funnel plot was not possible.

We decided to undertake subgroup analyses, as previously reported, and compare the effect of LL-BFRT in trials that used (i) $> 60\%$ 1RM load during exercise (high-load—HL) [6], or (ii) low intensity exercise ($<40\%$ of 1RM) [7, 30].

## Certainty of evidence

Two investigators (KP and GP) independently evaluated the certainty of evidence using the Grading of Recommendations, Assessment, Development and Evaluations (GRADE)

**Table 1. Criteria used for grading the certainty of evidence.**

| GRADE domain | Criteria for downgrading the certainty of evidence using the GRADE methodology |
|---|---|
| Risk of Bias | Certainty of evidence was downgraded one level, if the "low risk" studies contributed less than 50% of participants in the pairwise comparison (PEDro score <7 determined a study as having "high" risk of bias) |
| Inconsistency | Certainty of evidence was downgraded one level, if: (1) the overlap of 95%CIs presented in forest plots was poor; (2) the magnitude and direction of the effect was inconsistent between studies, and (3) the strength of the evidence suggesting substantial heterogeneity (p value from $\chi^2$ test, or $I^2$>50%) |
| Indirectness | Certainty of evidence was downgraded one level, if: heterogeneity in population characteristics or interventions was evident |
| Imprecision | Certainty of evidence was downgraded one level, if: (1) a sample size with adequate power for the outcome was not calculated and reported, and (2) the upper or lower 95%CI spanned an effect size of 0.5 in either direction |
| Publication bias | The presence of publication bias as assessed by funnel plots, where applicable. |

Abbreviations: CI, confidence intervals; GRADE, Grading of Recommendations Assessment, Development and Evaluation; PEDro, Physiotherapy Evidence Database.

methodology [31, 32] and created and exported tables using the gradepro software (https://gdt.gradepro.org/). Quality of the evidence was rated as "high", "moderate", "low" or "very low" depending on the presence of: risk of bias, inconsistency, indirectness, imprecision, publication bias (where applicable) (Table 1). Any disagreements were resolved by discussion and involvement of a third investigator (CK).

In the case of a single trial outcome, we a-priori graded the evidence as "low certainty", and if the study had a "high risk" of bias the evidence was downgraded to "very low certainty" [33].

## Results

### Study selection

The search strategy identified 346 potentially relevant publications, of which 23 articles were full text screened. Nine studies met the eligibility criteria and were included (Fig 1), of which eight [34–41] were RCTs and one [42] had a random quasi-experimental study design.

### Study characteristics and participants

Study characteristics such as, sample size, age, gender, interventions, intervention parameters, loading progressions, and main findings are presented in Table 2. All included studies recruited healthy participants (n = 218) of which the majority were male (73%) with a mean pooled age of 26.5 years (mean range 19.2 to 60.5 years). Five studies [37–41] recruited only male and one study [42] only female participants. The median number of participants randomized per trial was 24 (IQR 11–32) and the sample size ranged from 9 to 46.

### Intervention characteristics

The training duration ranged from 2 to 8 weeks (median = 3, IQR 2–3), with 2 to 8 exercise sessions per week (median = 2.5, IQR 2–3, range 2 to 6). Three studies [34–36] used only shoulder muscle exercises, two studies [37, 42] a combination of shoulder, chest, and back muscle exercises, while four [38–41] only the bench press exercise (Table 2). With regards to BFRT, all studies implemented 4 sets for each exercise with the same number of repetitions (75 total), except one study [43] that participants were instructed to perform the fourth set to

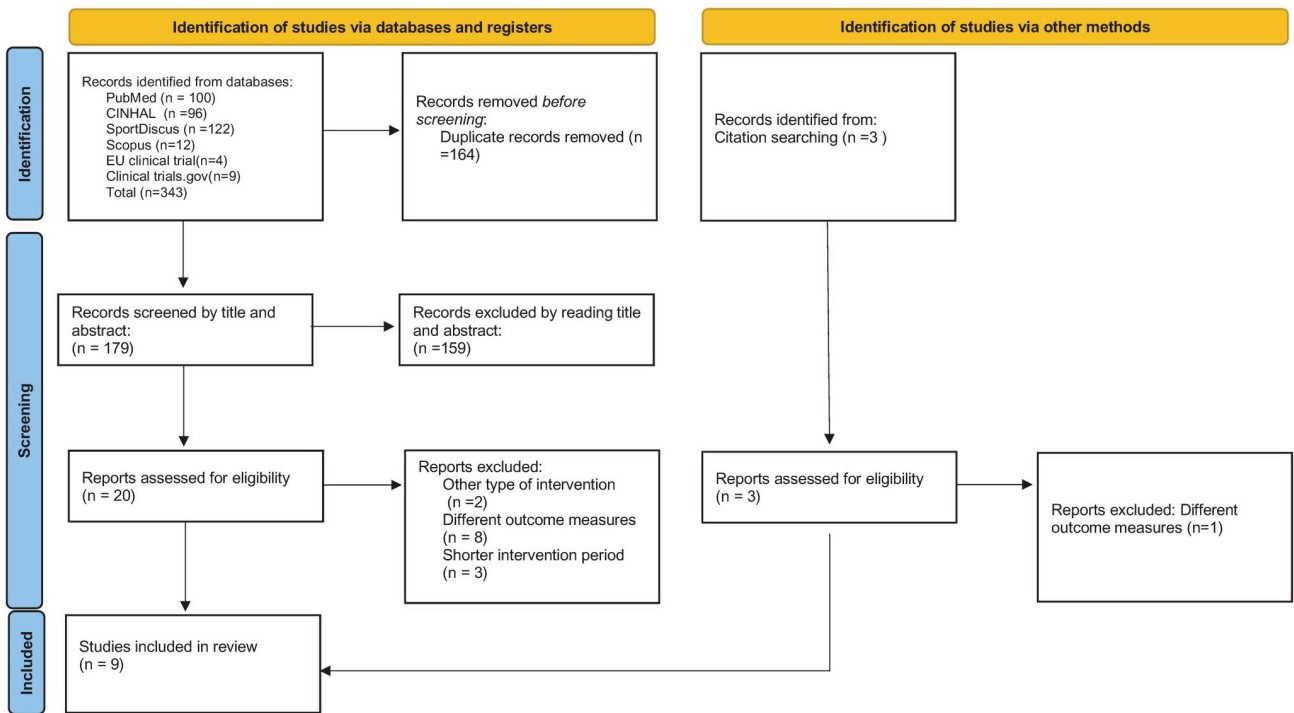

**Fig 1. The PRISMA flow diagram of the study selection process.**

volitional exhaustion and another study [39] that participants performed 100 repetitions in total. LL-RT and HL-RT groups used the same exercises with the BFRT group in every study, with a more variable protocol in terms of sets and repetitions; nevertheless, in 5 out of 9 included studies the comparator group shadowed the exercise volume of the BFRT group. Between-set rest ranged from 30 to 45 seconds and between-exercise rest from 30 secs to 2 minutes (Table 2). In all studies the training load was based on 1RM or maximum voluntary isometric contraction (MVIC). The BFRT training intensities were always low (10–30% of 1RM), while two studies gradually increased the training load from 10% to 30% and from 20% to 32% of the 1RM, respectively [38, 42]. The control group training intensities ranged from low intensity [34–36, 39, 40] to high intensity (65–90% of 1RM) [37, 38, 41, 42] (Table 2).

Four studies used automatically adjusted individualized cuff pressure ranging from 50% to 60% of the total vascular limb occlusion pressure (LOP) [34–37], two studies [38, 42] based the LOP to the RPE during exercising, while 4 of the included studies [38–41] applied an elastic band for the limb occlusion. Three studies [40–42] applied an incremental increase (weekly) of the occlusion pressure during the intervention based on predefined (not individualized) mmHg pressure (Table 2) [42]. Finally, one study [39] did not specify the used LOP.

## Quality assessment and risk of bias

Out of the 9 included studies, 8 (89%) had moderate methodological quality and received an overall "high risk" of bias rating (Table 3). The main methodological concerns were lack of therapist (9/9), assessor (7/9), and patient blinding (9/9), and unclear allocation concealment (9/9).

**Table 2. Participant and study characteristics, and physiological adaptations and main findings.**

| Article, year | Country | Participants characteristics (mean±SD age, activity level, % women) | Sample size), | Duration of intervention | Exercise(s) | BFRT group exercise parameters | Control group exercise parameters | Progression for BFRT group | Progression for Control group | Main findings in physiological adaptations in strength and muscle size |
|---|---|---|---|---|---|---|---|---|---|---|
| Bowman et al., 2020 | USA | total age:26.2 ±3.4 Trained young adults 58% women | BFR-RT, n = 14 LL-RT, n = 10 | 2 sessions per week for6 weeks | External rotation Internal rotation Biceps curl Triceps extension Prone horizontal abduction | 4 sets 30/15/15/15 repetitions 30 secs between-set rest 30% 1RM 60% LOP (Delfi system©) | Same training parameters without BFR | Weight was increased as needed to accomplish 7–8 RPE | Weight was increased as needed to accomplish 7–8 RPE | ↑30%, 23%, 22%, and 13% in scaption, flexion, abduction, and grip strength in favour of the BFRT group, respectively ↑arm and forearm circumferences in favour of the BFRT group (p<0.01) |
| Brumitt et al., 2020 | USA | total age:25.0 ±2.2 53% reported resistance training for the shoulders more than once per week 43.5% women | BFR-RT, n = 24 LL-RT, n = 22 | 2 sessions per week for 8 weeks | External rotation | 4 sets 30/15/15/15 repetitions 30 secs between-set rest 30% 1RM 50% LOP (Delfi system©) | Same training parameters without BFR | NA | NA | Significant within-group supraspinatus and external rotators strength gains for both BFR and LL-RT group (p<0.001) No between-group differences in supraspinatus (p = 0.750) and external rotator strength (p = 0.708) and supraspinatus tendon thickness (p = 0.610) |
| Green et al., 2020 | USA | range 20–29 years Trained young adults 0% women | BFR-RT, n = 6 HL-RT, n = 5 | 2 sessions per week for4 weeks | Bench press Scapular retraction External rotation Bent over row | 4 sets 30/15/15/15 repetitions Each exercise was performed within 7 min 1 minute rest between-exercise 20% 1RM 50% LOP (Delfi system©) | Same exercises without BFR 3 sets 10/10/10 repetitions 70% 1RM | NA | NA | No between-group significant strength differences (p>0.05) ↑within-BFRT strength for pectoralis major, lower trapezius, and 1RM for prone rows (p<0.05) ↑within-control group strength for pectoralis major, lower trapezius, external rotators, and 1RM for scapula retraction (p<0.05) |

(*Continued*)

**Table 2.** (Continued)

| Article, year | Country | Participants characteristics (mean±SD age, activity level, % women) | Sample size), | Duration of intervention | Exercise(s) | BFRT group exercise parameters | Control group exercise parameters | Progression for BFRT group | Progression for Control group | Main findings in physiological adaptations in strength and muscle size |
|---|---|---|---|---|---|---|---|---|---|---|
| Lambert et al., 2021 | USA | BFRT age:27.6 ±4.3 Control age:25.8±4.1 Untrained young adults 28% women | BFR-RT, n = 16 LL-RT, n = 16 | 2 sessions per week for 8 weeks | Internal rotation External rotation Side lying external rotation Scaption | 4 sets 30/15/15/fatigue repetitions 30 secs between-set rest 2 minutes rest between-exercise 20% of isometric max 50% LOP (Delfi system©) | Same training parameters without BFR | 1 lb per week per exercise if 75 reps achieved for both weekly sessions | 1 lb per week per exercise if 75 reps achieved for both weekly sessions | ↑shoulder region lean mass in the arm in favour of the BFRT group (p<0.05) ↑isometric strength (p<0.001) and strength endurance (p<0.01) for IR at 0˚ of ABD in favour of the BFRT group |
| Salyers, 2017 | USA | total age:22.1 ±1.5 Trained young adults 0% women | BFR-RT, n = 4 HL-RT, n = 4 | 3 sessions per week for 4 weeks | Bench press | 4 sets 30/15/15/15 repetitions 45 secs between-set rest 20% 1RM LOP: RPE 7/10 with elastic band | Same exercises without BFR 3 sets 15/15/15 repetitions 90 secs between-set rest Started with 65% 1RM | Weekly increased over 4 weeks 20% 1RM 25% 1RM 30% 1RM 32.5% 1RM | Weekly modified over 4 weeks 65% 1RM/15 repetitions 75% 1RM/10 repetitions 80% 1RM/8 repetitions 85% 1RM/6 repetitions | No significant between-group differences in body composition and strength measurements (p<0.05) |
| Thiebaud et al., 2013 | USA | BFRT age:59.0 ±2.0 Control age:62.0±2.0 Untrained older adults 100% women | BFR-RT, n = 6 HL-RT, n = 8 | 3 sessions per week for 8 weeks | Chest press Seated row Seated shoulder press | 3 sets 30/15/15 repetitions 30 secs between-set rest 30 secs to 2 minutes between-exercise rest 10–30% 1RM based on 7–9 RPE on OMNI-RES AM scale 80-120mmHg LOP (KAATSU-Master device©) | 3 sets 10/10/10 repetitions 2 min between-set rest 30 secs to 2 minutes between-exercise rest 70–90% 1RM based on 7–9 RPE on OMNI-RES AM scale | Weekly increased LOP over 4 weeks 80mmHg 90mmHg 100mmHg 110-120mmHg | Exercise modified to reach a 7–9 RPE on OMNI-RES AM scale | ↑within-group strength increases in chest press, seated row, shoulder press (p<0.05) and pectoralis major thickness in both groups (p<0.05) No between-group differences in lean body mass, strength, and muscle thickness (p>0.05) |

(*Continued*)

**Table 2.** (Continued)

| Article, year | Country | Participants characteristics (mean±SD age, activity level, % women) | Sample size), | Duration of intervention | Exercise(s) | BFRT group exercise parameters | Control group exercise parameters | Progression for BFRT group | Progression for Control group | Main findings in physiological adaptations in strength and muscle size |
|---|---|---|---|---|---|---|---|---|---|---|
| Yamanaka et al., 2012 | USA | total age 19.2 ±1.8, Trained young adults 0% women | BFR-RT, n = 16 LL-RT, n = 16 | 3 sessions per week for 4 weeks | Bench press | 4 sets 30/20/20/20 repetitions 45 secs between-set rest 20% 1RM LOP restricted by elastic band (pulled to overlap 2 inches) | Same training parameters without BFR | NA | NA | ↑within-group strength and girth measures for both groups (p<0.05) ↑bench press 1RM (7% within-group), upper and lower chest girths (within-group 3% and 3%, respectively), and left upper arm girth in favour of the BFRT group (p<0.05) |
| Yasuda et al., 2010 | Japan | BFRT age:25.8 ±6.3 Control age:25.6±3.2 Trained young adults (no resistance training for a year) 0% women | BFR-RT, n = 5 LL-RT, n = 5 | 6 sessions per week for 2 weeks | Bench press | 4 sets 30/15/15/15 repetitions 30 secs between-set rest 30% 1RM LOP using an elastic band: The training air pressure started at 100mmHG | Same training parameters without BFR | Training pressure started at 100mmHG and was increased by 10 mmHg each day until 160 mmHg (Day 7) | NA | Significant increase in triceps brachii (8%) and pectoralis major muscle (16%) thickness in favour of the BFRT group (p<0.05) compared to the non-BFRT group Significant increase (p<0.05) in 1-RM bench press strength (6%) in favour of the BFR-T (6%) group compared to the non-BFRT group |

(*Continued*)

**Table 2.** (Continued)

| Article, year | Country | Participants characteristics (mean±SD age, activity level, % women) | Sample size), | Duration of intervention | Exercise(s) | BFRT group exercise parameters | Control group exercise parameters | Progression for BFRT group | Progression for Control group | Main findings in physiological adaptations in strength and muscle size |
|---|---|---|---|---|---|---|---|---|---|---|
| Yasuda et al., 2011 | Japan | HI-RT age:25.3 ±2.9 BFRT$_L$ age:23.4 ±1.3 BFRT$_C$ age:23.8 ±2.1 Control age:23.6±1.6 Trained young adults (no resistance training for 6 months) 0% women | HI-RT, n = 10 BFRT$_L$, n = 10 BFRT$_C$, n = 10 Control, n = 10 | 3 sessions per week for 8 weeks | Bench press | 4 sets 30/15/15/15 repetitions 30 secs between-set rest 30% 1RM LOP elastic band: The training air pressure started at 100mmHG and was increased by 10 mmHg each day until 160 mmHg (Day 7) | HI-RT: 3 sets 10/10/10 repetitions 2–3 minutes between-set rest 75% 1RM BFRT$_C$: performed BFRT$_L$ twice a week and HI-RT once a week | 1-RM was assessed after 3 weeks to adjust the training load for BFRT$_H$. Training load was constant for BFRT$_L$ | NA | Similar increases in bench press 1-RM in the HI-RT (19.9%) and BFRT$_C$ (15.3%) groups and lower in the BFRT$_L$ group (8.7%, p<0.05) ↑11.3% and 6.6% in maximal isometric elbow extension for BFRT$_H$ and BFRT$_C$, respectively ↑8.6%, 7.2%, and 4.4% in the cross-sectional area of the triceps brachii for HI-RT, BFRT$_C$, and BFRT$_L$, respectively Significant change in relative isometric strength (3.3%) in favour of HI-RT (p<0.05) compared to BFRT$_L$ (-3.5%) and control (-0.1%) groups |

Abbreviations: ABD, abduction; BFRT, blood flow restriction training; C, combined high intensity resistance training and low BFRT; HI-RT, high intensity resistance training; IR, internal rotation; L, low; LOP, limb occlusion pressure; NA, not applicable; OMNI-RES AM, OMNI Resistance for active muscle scale; RM, repetition maximum; RPE, rating of perceived exertion; SD, standard deviation; secs, seconds.

## Outcome measures

Dynamic muscle strength (1RM or using isokinetic dynamometry) was measured in seven studies [34, 37–42] and isometric muscle strength by using hand-held dynamometer in four studies (S2 Table in S1 File) [34–37].

A diversity of measurements (S2 Table in S1 File) was used to evaluate and quantify muscle size, including ultrasound (US) [35, 40, 42], or magnetic resonance imaging (MRI) [41] for the measurement of the cross-sectional area (CSA) of the muscles, dual energy X-ray absorptiometry (DEXA) for the lean body mass [36], and tape for measuring the circumference of the limb [34, 38, 39]. The measurement of tendon volume was conducted only in one study [35] using US.

**Table 3. Methodological quality and risk of bias of the included studies (PEDro scale).**

| PEDro scale | Brumit et al., 2020 | Bowman et al., 2020 | Lambert. et al., 2021 | Yasuda et al., 2011 | Yasuda et al., 2010 | Yamanaka et al., 2012 | Thiebaud et al., 2013 | Green et al., 2020 | Salyers, 2017 | Percent (%) |
|---|---|---|---|---|---|---|---|---|---|---|
| Eligibility criteria specified | Yes | Yes | Yes | No | No | Yes | Yes | No | No | 55.5% |
| Random allocation | Yes | Yes | Yes | Yes | Yes | Yes | No | Yes | Yes | 89% |
| Concealed allocation | No | No | No | No | No | No | No | No | No | 0% |
| Baseline comparability | No | Yes | Yes | Yes | Yes | Yes | Yes | No | No | 66.7% |
| Participant blinding | No | No | No | No | No | No | No | No | No | 0% |
| Therapist blinding | No | No | No | No | No | No | No | No | No | 0% |
| Assessor blinding | Yes | Yes | No | No | No | No | No | No | No | 22% |
| Adequate follow-up | Yes | Yes | Yes | Yes | Yes | Yes | Yes | Yes | Yes | 100% |
| Intention-to-treat analysis | No | Yes | No | Yes | Yes | Yes | No | Yes | Yes | 66.7% |
| Between-group comparisons | Yes | Yes | Yes | Yes | Yes | Yes | Yes | Yes | Yes | 100% |
| Point estimates & variability | Yes | Yes | Yes | Yes | Yes | Yes | Yes | Yes | Yes | 100% |
| Total PEDro score (Risk of Bias) | 6/10 (High risk) | 8/10 (Low risk) | 6/10 (High risk) | 6/10 (High risk) | 6/10 (High risk) | 6/10 (High risk) | 5/10 (High risk) | 5/10 (High risk) | 5/10 (High risk) | |

## Adverse events

Only one study [34] assessed adverse events and none were reported.

## Effects of interventions in muscle strength

**Muscle strength in LL-BFRT compared to LL-RT without BFR.** Five studies [34–36, 39, 40] evaluated the effect of LL-BFRT compared to LL-RT, of which only one was of low risk of bias [34] in a range of muscle movements using isokinetic or hand-held dynamometry (S2 Table in S1 File).

Very low certainty evidence suggests a significant improvement in bench press 1RM (SMD = 0.87) [39, 40] and shoulder flexion strength (SMD = 1.64) [34, 36] in favor of the LL-BFRT group (Table 4, Fig 2).

Three studies [34–36] evaluated shoulder external rotation strength and reported inconsistent between-group results affected by testing position and measurement method. Pooled results from two studies [35, 36] showed very low certainty evidence of no significant differences between comparators in shoulder isometric external rotation at 90° of abduction in prone strength but significant and unexplained heterogeneity (SMD = 0.96, $I^2$ = 92%) (S1a Fig in S1 File). Inconsistent results of low and very low-quality evidence were presented for shoulder external rotation strength in seated and in prone position; however, both studies [34, 36] that evaluated strength in seated position did not report significant between-group differences (Table 4).

Two studies [34, 36] evaluated shoulder internal rotation using different settings and reported low and very low certainty evidence of inconsistent results in individual-study calculated effect sizes at the 8-week follow-up (Table 4).

Shoulder abduction [34, 35] and scaption [34, 36] were evaluated in studies potentially using different measurement methodology (lack of information in one [34] and the evidence should be interpreted with caution. Pooled results for each outcome presented substantial

**Table 4. Summary of evidence for the effects of LL-BFRT compared with LL-, HL-RT, or no exercise in muscle strength.**

| Outcome—Strength | Comparisons | | Relative effect (95%CI) | BFRT / comparator (n studies) | Quality of evidence (GRADE) | Evidence and significance |
|---|---|---|---|---|---|---|
| | Average estimate in BFRT group | Average estimate in comparator group | | | | |
| **Bench press–1RM** | **LL-BFRT:** | **LL-RT:** | **SMD 0.87** | 21/21 | ⊕◯◯◯ | Very low certainty evidence of a significant difference in bench press strength (1RM in Kgs) in favor of LL-BFRT compared to LL-RT at 2–4 weeks follow-up |
| Follow-up 2–4 weeks | Pooled weighted mean±SD was 119.8 ±15.4 (mean range 62.0 to 137.9) | Pooled weighted mean ±SD was 105.0±16.4 (mean range 57.5 to 119.8) | **[0.23, 1.51]** Statistically significant difference | (2) | **Very low**[1,3,4] | |
| **Flexion—MVIC** | **LL-BFRT:** | **LL-RT:** | **SMD 1.64** | 30/26 | ⊕◯◯◯ | Very low certainty evidence of a significant difference in shoulder flexion strength (MVIC) in favor of the LL-BFRT group at the 6–8 weeks follow-up |
| Follow-up 6–8 weeks | Pooled weighted mean±SD was 14.15 ±1.16 (mean range 12.8 to 15.7) | Pooled weighted mean ±SD was 10.0±0.64 (mean range 6.97 to 11.9) | **[0.57, 2.71]** Statistically significant difference | (2) | **Very low**[1,3,4] | |
| **External rotation in prone 90° – MVIC** | **LL-BFRT:** | **LL-RT:** | **MD 1.40** | 16/16 | ⊕◯◯◯ | Very low certainty evidence of a significant difference in external rotation in prone at 90° of abduction strength in favor of the LL-BFRT at the 8-week follow-up |
| Follow-up 8 weeks | Mean±SD was 16.1 ±0.7 | Mean±SD was 14.7±0.7 | **[0.91, 1.89]** Statistically significant difference | (1) | **Very low**[1] | |
| **External rotation in prone 90° – MVIC** | **LL-BFRT:** | **LL-RT:** | **MD 0.19** | 24/22 | ⊕◯◯◯ | Very low certainty evidence of no significant difference in external rotation in prone at 90° of abduction strength between the comparators at the 8-week follow-up |
| Follow-up 8 weeks | Mean±SD was 13.54 ±4.94 | Mean±SD was 13.35 ±5.7 | **[-2.91, 3.29]** Non statistically significant difference | (1) | **Very low**[1] | |
| **External rotation seated —Peak torque Nm** | **LL-BFRT:** | **LL-RT:** | **MD 5.30** **[-0.18, 10.78]** | 14/10 | ⊕⊕◯◯ | Low certainty evidence of no significant difference in shoulder external rotation (in seated position) peak torque (isokinetic dynamometry) between LL-BFRT and LL-RT at the 6-week follow-up |
| Follow-up 6 weeks | Mean±SD was 20.4 ±6.7 | Mean±SD was 15.1±6.8 | Non statistically significant difference | (1) | **Low** | |
| **External rotation seated —MVIC** | **LL-BFRT:** | **LL-RT:** | **MD 0.20** | 16/16 | ⊕◯◯◯ | Low certainty evidence of no significant difference in shoulder external rotation MVIC (in seated position) between LL-BFRT and LL-RT at the 8-week follow-up |
| Follow-up 8 weeks | Mean±SD was 20.4 ±6.7 | Mean±SD was 15.1±6.8 | **[-0.08, 0.48]** Non statistically significant difference | (1) | **Very low**[1] | |
| **Internal rotation seated —MVIC** | **LL-BFRT:** | **LL-RT:** | **MD 2.90** | 16/16 | ⊕◯◯◯ | Very low certainty evidence of a significant difference in (seated) internal rotation at 0° of abduction strength in favor of the LL-BFRT at the 8-week follow-up |
| Follow-up 8 weeks | Mean±SD was 23.1 ±0.7 | Mean±SD was 13.9±0.4 | **[2.41, 3.39]** Statistically significant difference | (1) | **Very low**[1] | |

*(Continued)*

**Table 4.** (Continued)

| Outcome—Strength | Comparisons | | Relative effect (95%CI) | BFRT / comparator (n studies) | Quality of evidence (GRADE) | Evidence and significance |
|---|---|---|---|---|---|---|
| | Average estimate in BFRT group | Average estimate in comparator group | | | | |
| **Internal rotation in prone 90º – MVIC** | **LL-BFRT:** | **LL-RT:** | **MD 0.50** | 16/16 | ⊕○○○ | Very low certainty evidence of no significant difference in prone internal rotation at 90° of abduction strength between LL-BFRT and LL-RT at the 8-week follow-up |
| Follow-up 8 weeks | Mean±SD was 18.6 ±0.8 | Mean±SD was 18.1±0.8 | [-0.05, 1.05] | (1) | **Very low[1]** | |
| | | | Non statistically significant difference | | | |
| **Abduction—MVIC** | **LL-BFRT:** | **LL-RT:** | **MD 3.85** | 14/10 | ⊕⊕○○ | Low certainty evidence of a significant difference in shoulder abduction strength in favor of the LL-BFRT group at the 6-week follow-up |
| Follow-up 6 weeks | Mean±SD was 8.6 ±4.1 | Mean±SD was 4.7±3.1 | [0.95, 6.75] | (1) | **Low** | |
| | | | Statistically significant difference | | | |
| **Abduction—MVIC** | **LL-BFRT:** | **LL-RT:** | **MD -0.41** | 24/22 | ⊕○○○ | Very low certainty evidence of no significant difference in shoulder abduction strength between LL-BFRT and LL-RT at the 8-week follow-up |
| Follow-up 8 weeks | Mean±SD was 18.9 ±3.5 | Mean±SD was 19.3±4.1 | [-2.61, 1.79] | (1) | **Very low[1]** | |
| | | | Non statistically significant difference | | | |
| **Scaption—MVIC** | **LL-BFRT:** | **LL-RT:** | **MD 4.77** | 14/10 | ⊕⊕○○ | Low certainty evidence of a significant difference in shoulder scaption strength in favor of the LL-BFRT group at the 6-week follow-up |
| Follow-up 6 weeks | Mean±SD was 38.9 ±18.2 | Mean±SD was 22.3 ±13.7 | [1.64, 7.90] | (1) | **Low** | |
| | | | Statistically significant difference | | | |
| **Scaption—MVIC** | **LL-BFRT:** | **LL-RT:** | **MD -0.10** | 16/16 | ⊕○○○ | Very low certainty evidence of no significant difference in shoulder scaption strength between LL-BFRT and LL-RT at the 8-week follow-up |
| Follow-up 8 weeks | Mean±SD was 12.4 ±0.2 | Mean±SD was 12.5±0.2 | [-0.24, 0.04] | (1) | **Very low[1]** | |
| | | | Non statistically significant difference | | | |
| **Strength (Bench press—1RM)** Follow-up 4–8 weeks | **LL-BFRT:** Pooled weighted mean±SD was 56.4 ±9.7 (mean range 53.7 to 105.2) | **HL-RT:** Pooled weighted mean ±SD was 55.3±10.0 (mean range 56.78 to 102.8) | **SMD -0.17** [-0.78, 0.44] Non statistically significant difference | 20/22 (3) | ⊕○○○ **Very low[1,3,4]** | Very low certainty evidence of no significant difference in bench press 1RM strength between LL-BFRT and HL-RT at the 4–8 weeks follow-up |
| **Strength (Seated row–1RM)** Follow-up 8 weeks | **LL-BFRT:** Mean±SD was 39.8 ±3.2 | **HL-RT:** Mean±SD was 40.1±5.1 | **MD -0.31** [-4.68, 4.06] Non statistically significant difference | 6/8 (1) | ⊕○○○ **Very low[1]** | Very low certainty evidence of no significant difference in seated row strength between LL-BFRT and HL-RT at the 8-week follow-up |
| **Strength (Shoulder press –1RM)** Follow-up 8 weeks | **LL-BFRT:** Mean±SD was 21.3 ±5.1 | **HL-RT:** Mean±SD was 19.6±4.8 | **MD 1.69** [-3.56, 6.94] Non statistically significant difference | 6/8 (1) | ⊕○○○ **Very low[1]** | Very low certainty evidence of no significant difference in shoulder press strength between LL-BFRT and HL-RT at the 8-week follow-up |

(Continued)

**Table 4.** (Continued)

| Outcome—Strength | Comparisons | | Relative effect (95%CI) | BFRT / comparator (n studies) | Quality of evidence (GRADE) | Evidence and significance |
|---|---|---|---|---|---|---|
| | Average estimate in BFRT group | Average estimate in comparator group | | | | |
| **Pectoralis major —MVIC** | **LL-BFRT:** | **HL-RT:** | **MD -14.32** | 6/5 | ⊕◯◯◯ | Very low certainty evidence of a significant difference in pectoralis major in favor of HL-RT at the 4-week follow-up |
| Follow-up 4 weeks | Mean±SD was 49.6 ±8.1 | Mean±SD was 63.9±7.1 | **[-23.35, -5.29]** Statistically significant difference | (1) | **Very low**[1] | |
| **Lower trapezius —MVIC** | **LL-BFRT:** | **HL-RT:** | **MD -4.92** | 6/5 | ⊕◯◯◯ | Very low certainty evidence of no significant difference in lower trapezius strength between LL-BFRT and HL-RT at the 4-week follow-up |
| Follow-up 4 weeks | Mean±SD was 32.6 ±5.5 | Mean±SD was 37.5±4.6 | **[-10.90, 1.06]** Non statistically significant difference | (1) | **Very low**[1] | |
| **External rotation in prone 90º – MVIC** | **LL-BFRT:** | **HL-RT:** | **MD -2.72** | 6/5 | ⊕◯◯◯ | Very low certainty evidence of no significant difference in external rotation in prone and 90˚ of shoulder abduction strength between LL-BFRT and HL-RT at the 4-week follow-up |
| Follow-up 4 weeks | Mean±SD was 39.6 ±15.8 | Mean±SD was 42.3±9.2 | **[-17.71, 12.27]** Non statistically significant difference | (1) | **Very low**[1] | |
| **Prone row— MVIC** | **LL-BFRT:** Mean±SD was | **HL-RT:** Mean±SD was | **MD -0.83** **[-21.60, 19.94]** | 6/5 | ⊕◯◯◯ | Very low certainty evidence of no significant difference in prone row strength between LL-BFRT and HL-RT at the 4-week follow-up |
| Follow-up 4 weeks | 104.2±18.6 | 105.0±16.6 | Non statistically significant difference | (1) | **Very low**[1] | |
| **Scapular retraction— MVIC** | **LL-BFRT:** | **HL-RT:** | **MD 0.50** | 6/5 | ⊕◯◯◯ | Very low certainty evidence of no significant difference in scapular retraction strength between LL-BFRT and HL-RT at the 4-week follow-up |
| Follow-up 4 weeks | Mean±SD was 247.5 ±34.2 | Mean±SD was 247.0 ±25.9 | **[-35.03, 36.03]** Non statistically significant difference | (1) | **Very low**[1] | |

[1] Downgraded due to Risk of Bias.

[2] Downgraded due to inconsistency.

[3] Downgraded due to indirectness.

[4] Downgraded due to imprecision.

Abbreviations: GRADE, Grading of Recommendations Assessment, Development and Evaluation; LL, low load; LL-BFRT, low load blood flow restriction training; HL, high load; MD, mean difference; MVIC, maximum voluntary isometric contraction; 1RM, one repletion maximum; RT, resistance training; SD, standard deviation; SMD, standardised mean difference.

heterogeneity ($I^2 > 75\%$), thus we did not perform a formal meta-analysis (S1b and S1c Fig in S1 File). By calculating MDs for each study, inconsistent results showed low certainty evidence in favor of LL-BRT in abduction and scaption, strength of the shoulder from one study [34] and very low certainty evidence of no difference between the comparators in abduction and scaption shoulder strength from another study [36] (Table 4).

**Muscle strength in LL-BFRT compared to HL-RT.** Four studies [37, 38, 41, 42] with high risk of bias evaluated the effects of LL-BFR training compared to HL-RT (>60 1RM) in muscle strength in a wide range of exercises.

a

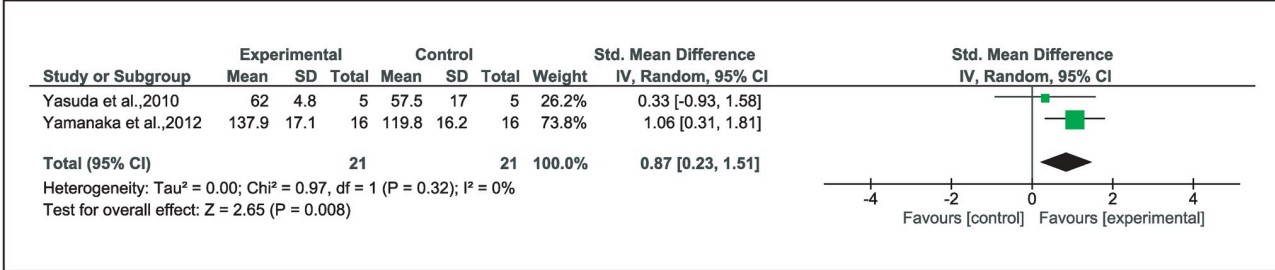

b

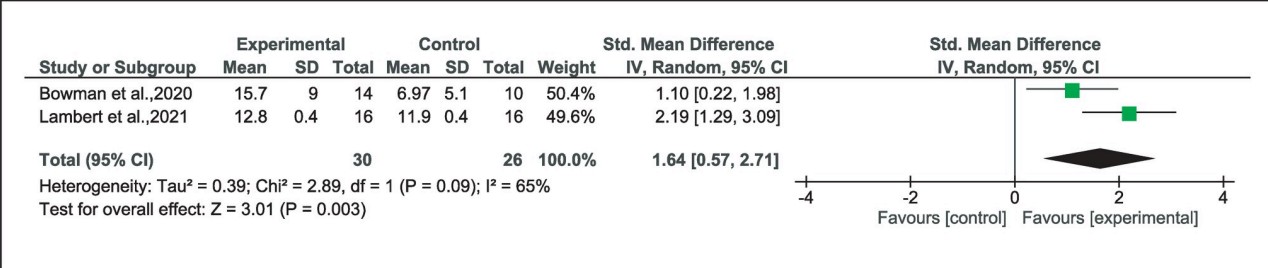

**Fig 2. Forest plots depicting studies using LL-BFRT compared to studies using LL-RT in muscle strength.** Forest plot comparing low-load resistance training with blood flow restriction (LL-BFR) and low-load resistance training alone (LL-RT) on muscle strength of a) chest press (1RM) b) shoulder flexion (dynamometry in kgs). Abbreviations: CI, confidence interval; IV, inverse variance; Random, random effects model; SE, standard error.

Very low certainty evidence from three studies [38, 41, 42] suggested no significant difference between the comparators in bench press 1RM (SMD = -0.17) (Fig 3); however, very low certainty of evidence from one study [37] suggested a significant difference in pectoralis major MVIC in favor of HL-RT (MD = -14.32) (Table 4). Exclusion of one study including a significantly older population [42] did not change the certainty of evidence, the direction, and the size (SMD = -0.15, 95%CI: -0.90 to 0.59) of the effect estimate (Fig 3). Additionally, very low certainty evidence [42] indicates no significant differences between LL-BFR training and HL-RT training in 1RM shoulder press and seated row exercise (Table 4).

Finally, from a study [42] assessing isometric strength of several upper body muscles in senior individuals (age >59 years), very low certainty evidence indicates no significant differences between LL-BFRT and HL-RT in lower trapezius, external rotation, prone row, seated row, and scapular retraction strength at the 8-week follow-up (Table 4).

## Effects of interventions in muscle size

**Muscle size in LL-BFRT compared to LL-RT without BFRT.** Three studies [36, 39, 40] evaluated the effect of LL-BFRT training compared to LL-RT in the size of muscles located proximally to the application of the BFRT by using US, DEXA, or tape measure (S2 Table in S1 File).

Very low certainty evidence suggests a significant increase in shoulder lean mass measured with DEXA in favor of the LL-BFRT group compared to the LL-RT group at the 8-week follow-up (Table 5). Very low certainty evidence of no significant differences between comparators were reported for pectoralis major thickness, and upper and lower chest girth [39, 40] (Table 5).

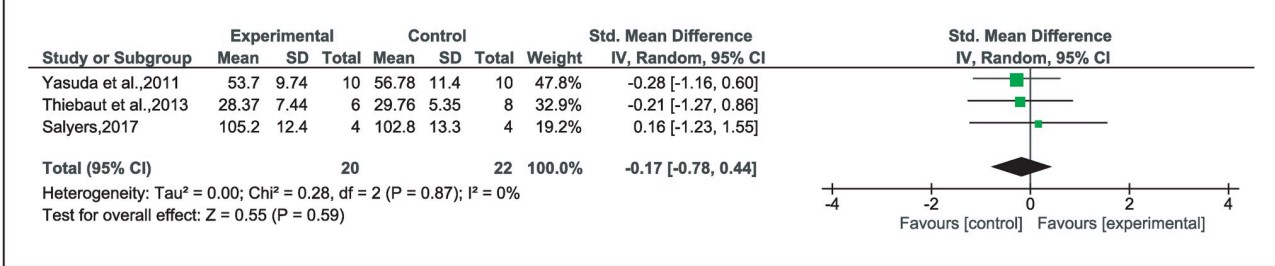

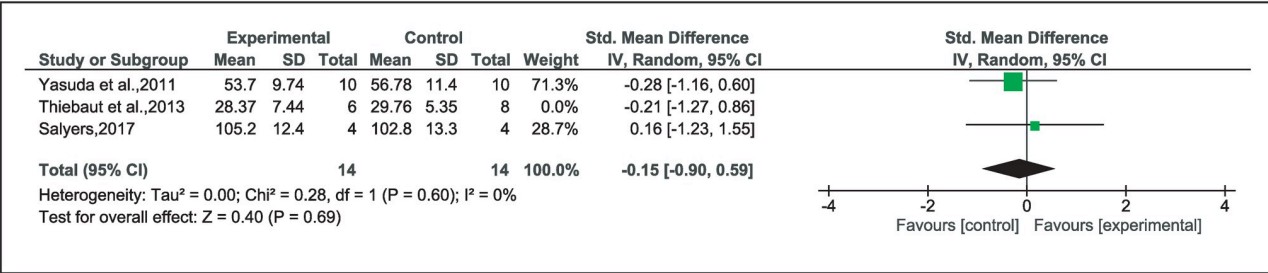

**Fig 3. Forest plots depicting studies using LL-BFRT compared to studies using HL-RT in muscle strength.** Forest plots comparing low-load resistance training with blood flow restriction (LL-BFRT) and high load resistance training alone HL-RT in strength of: a) chest press (1RM), and b) sensitivity analysis by removing one study (Thiebaud et al., 2013) that included substantially older participants. Abbreviations: CI, confidence interval; IV, inverse variance; Random, random effects model; SD, standard deviation.

**Muscle size in LL-BFRT compared to HL-RT without BFR.** Three studies [38, 41, 42] evaluated the effect of LL-BFRT compared to HL-RT in the size of muscles located proximally to the application of the BFR by using US, DEXA, MRI, or tape measure (S2 Table in S1 File).

Very low certainty evidence [42] suggests a significant increase in pectoralis major thickness (MD = 0.57) in favor of the LL-BFRT group compared to the HL-RT group (Table 5). Very low certainty evidence showed no significant differences between the comparators in deltoid muscle thickness, lean trunk mass, pectoralis muscle CSA, or chest girth (Table 5).

### Effects of interventions in tendon thickness

**Tendon thickness in LL-BFRT compared to LL-RT without BFR.** Only one study with high risk of bias [35] evaluated the effect of LL-BFRT compared to LL-RT without BFR and reported very low certainty evidence of no significant differences between the comparators in supraspinatus tendon thickness (MD = -0.01 95%CI: -0.05 to 0.02) (Table 5).

## Discussion

The findings of this systematic review suggest that LL-BFRT may result in better strength and size adaptations compared to similar exercise without BFR in muscles proximal to the applied cuff, although the quality of evidence is low, and the findings are mixed. Low and very low certainty evidence suggests a significant increase in bench press 1RM (2–4 weeks) and in shoulder flexion MVIC (6–8 weeks) in favor of the LL-BFRT compared to the LL-RT without BFR group. Conflicting evidence (low and very low certainty) was found for shoulder abduction, scaption, internal and external rotation strength. Very low certainty evidence suggests a significant increase in pectoralis major MVIC in favor of the HL-RT compared to the LL-BFR group

**Table 5. Summary of evidence for the effects of LL-BFRT compared with LL-, HL-RT, or no exercise in muscle size.**

| Outcome—Muscle size | Comparisons | | Relative effect (95%CI) | BFRT / comparator (n studies) | Quality of evidence (GRADE) | Evidence and significance |
|---|---|---|---|---|---|---|
| | Average estimate in BFRT group | Average estimate in comparator group | | | | |
| **Pectoralis major—thickness US (cm)** | LL-BFRT: | LL-RT: | MD 0.54 | 5/5 | ⊕◯◯◯ | Very low certainty evidence of no significant difference in pectoralis major thickness (cm) between LL-BFRT and LL-RT at the 2-week follow-up |
| Follow-up 2 weeks | Mean±SD was 2.76±2.0 | Mean±SD was 2.22 ±4.9 | [-4.14, 5.22] Non statistically significant difference | (1) | **Very low**[1] | |
| **Upper chest—girth (cm)** | LL-BFRT: | LL-RT: | MD 3.20 | 16/16 | ⊕◯◯◯ | Very low certainty evidence of no significant difference in upper chest girth between LL-BFRT and LL-RT at the 8-week follow-up |
| Follow-up 8 weeks | Mean±SD was 112.3±5.8 | Mean±SD was 109.1±5.1 | [-0.58, 6.98] Non statistically significant difference | (1) | **Very low**[1] | |
| **Lower chest—girth (cm)** | LL-BFRT: | LL-RT: | MD 1.20 [-2.14, 4.54] | 16/16 | ⊕◯◯◯ | Very low certainty evidence of no significant difference in lower chest girth between LL-BFRT and LL-RT at the 8-week follow-up |
| Follow-up 8 weeks | Mean±SD was 102.3±4.4 | Mean±SD was 101.1±5.2 | Non statistically significant difference | (1) | **Very low**[1] | |
| **Shoulder lean mass—DEXA (Kg)** | LL-BFRT: | LL-RT: | MD 0.18 | 16/16 | ⊕◯◯◯ | Very low certainty evidence of a significant increase in shoulder lean mass (DEXA) in favor of the LL-BFRT group at the 8-week follow-up |
| Follow-up 8 weeks | Mean±SD was 0.278±0.09 | Mean±SD was 0.096±0.061 | [0.13, 0.24] Statistically significant difference | (1) | **Very low**[1] | |
| **Pectoralis major—thickness US (cm)** | LL-BFRT: | HL-RT: | MD 0.57 | 6/8 | ⊕◯◯◯ | Very low certainty evidence of a significant increase in pectoralis major thickness in cm in favor of the LL-BFRT group compared to the HL-RT group at the 8-week follow-up |
| Follow-up 8 weeks | Mean±SD was 2.9±0.47 | Mean±SD was 2.33 ±0.46 | [0.08, 1.06] Statistically significant difference | (1) | **Very low**[1] | |
| **Deltoid—thickness US (cm)** | LL-BFRT: | HL-RT: | MD 0.12 | 6/8 | ⊕◯◯◯ | Very low certainty evidence of no significant difference in deltoid thickness (cm) between LL-BFRT and HL-RT at the 8-week follow-up |
| Follow-up 8 weeks | Mean±SD was 2.75±0.39 | Mean±SD was 2.63 ±0.46 | [-0.33, 0.57] Non statistically significant difference | (1) | **Very low**[1] | |
| **Trunk lean mass—DEXA (Kg)** | LL-BFRT: | HL-RT: | MD -1.00 | 6/8 | ⊕◯◯◯ | Very low certainty evidence of no significant difference in trunk lean mass (DEXA—Kg) between LL-BFRT and HL-RT at the 8-week follow-up |
| Follow-up 8 weeks | Mean±SD was 20.7±2.5 | Mean±SD was 21.7 ±4.1 | [-4.47, 2.47] Non statistically significant difference | (1) | **Very low**[1] | |

*(Continued)*

**Table 5.** (Continued)

| Outcome—Muscle size | Comparisons | | Relative effect (95%CI) | BFRT / comparator (n studies) | Quality of evidence (GRADE) | Evidence and significance |
|---|---|---|---|---|---|---|
| | Average estimate in BFRT group | Average estimate in comparator group | | | | |
| **Pectoralis major —CSA MRI (cm$^2$)** | **LL-BFRT:** | **HL-RT:** | **MD 1.10** | 10/10 | ⊕◯◯◯ | Very low certainty evidence of no significant difference in pectoralis major CSA between LL-BFRT and HL-RT at the 8-week follow-up |
| Follow-up 8 weeks | Mean±SD was 34.5±5.6 | Mean±SD was 33.4 ±6.9 | [-4.41, 6.61] Non statistically significant difference | (1) | **Very low**[1] | |
| **Chest—girth (cm)** | **LL-BFRT:** | **HL-RT:** | **MD -1.50** | 4/4 | ⊕◯◯◯ | Very low certainty evidence of no significant difference in chest girth between LL-BFRT and HL-RT at the 4-week follow-up |
| Follow-up 4 weeks | Mean±SD was 93.2±9.6 | Mean±SD was 94.7 ±13.2 | [-17.51, 14.51] Non statistically significant difference | (1) | **Very low**[1] | |

[1] Downgraded due to Risk of Bias.

[2] Downgraded due to inconsistency.

[3] Downgraded due to indirectness.

[4] Downgraded due to imprecision.

Abbreviations: CSA, cross sectional area; DEXA, Dual energy X-ray absorptiometry; GRADE, Grading of Recommendations Assessment, Development and Evaluation;: HL, high load; LL-BFRT, low load blood flow restriction training; LL, low load; MD, mean difference; MRI, magnetic resonance imaging; MVIC, maximum voluntary isometric contraction;1RM, one repletion maximum; RT, resistance training; SD, standard deviation; SMD, standardised mean difference; US, ultrasound.

at the 4-week follow-up, but evidence (low and very low certainty) of no significant differences in back and shoulder muscles were found between the comparators.

Very low certainty evidence suggests no significant differences in measures of chest muscles' size between LL-BFRT and LL-RT without BFR; however, very low certainty evidence indicates a significant effect of LL-BFRT in shoulder lean mass at the 8-week follow-up. In addition, very low certainty evidence indicates conflicting (pectoralis major) or non-significant (deltoid, chest girth, and trunk lean mass) differences between LL-BFRT and HL-RT in muscle size.

## Effect of LL-BFRT in muscle strength

Consistent evidence suggests that LL-BFRT induces larger improvements in muscle strength when compared—in both the upper and the lower limb—to LL-RT in young [19, 44] and older (>50 years) healthy individuals [30, 45, 46]. The size of the effect appears to be associated with the age of the healthy participants, the training duration, and the volume of exercise loading [19, 44–47] however, the measured effect was in muscles distal to the site of applied occlusive pressure of the exercised limb. Contrary to the notion that muscles not exposed to the restrictive stimulus would not have any benefit from the application of BFR, the studies included in the present review provide preliminary evidence for the opposite finding. It has been argued that exercising until volitional exhaustion of the prime mover muscles will increase the required activation of the synergistic muscles involved in the performance of an exercise (e.g., the synergistic involvement of the triceps muscle in the performance of a chest press exercise) and that this increase in activation may elicit increases in strength of muscles located proximal to the BFR cuff [15]. The authors hypothesized that the increasing fatiguing

effect on the triceps muscle under BFR would have caused a greater stress and demand on the pectoralis major muscle to compensate for the loss of force production [15]. Studies on the lower limb have shown that LL-BFRT may elicit such adaptations as indicated by an increase in (proximal) hip abductors strength [22] and of the gluteus maximus [43, 48]. Several mechanisms of BFR action have been proposed with the metabolic stress upregulating distinct cellular signaling pathways, along with the cell swelling in the hypoxic environment under BFRT being the most popular [4, 49–51]. It has been suggested that the intracellular swelling may serve as a stimulus to promote protein synthesis and inhibit proteolysis [22] and it seems that this mechanism is plausible in inducing proximal adaptations as a greater increase in swelling of the chest muscles was observed compared with the triceps at the end of low-load BFR bench press exercises [41].

In the present systematic review, we suggest that the significant between-group differences in muscle strength in favor of the LL-BFRT should be interpreted with caution given the unstandardized protocols with unmatched exercise loading and the diversity of outcome measurement methods. To illustrate, the two studies [34, 36] that showed significant improvement in shoulder flexion strength between LL-BFRT and LL-RT: a) varied significantly in the loading parameters used for the LL-BFR group (30% 1RM vs 20% of isometric max), b) used different load progressions over 6 and 8 weeks (load increase by 1 lb per week if 75 repetitions were achieved [36] versus load increase based on a score >7/10 on RPE scale [34], c) had unbalanced training volume (i.e., the fourth set performed to fatigue [36], and d) implemented different exercise programs (one study [34] did not include shoulder flexion exercise at all). Furthermore, significant methodological differences could be noted between the two studies reported favorable outcomes for the LL-BFRT in the bench press 1RM [39, 40]. Both studies occluded both upper arms simultaneously (in contrast to the majority of the included studies), while the exercise protocol for the study by Yamanaka et al. [39] involved parallel BFRT of the upper and lower limb showing a significant imbalance in the total time under occlusion, and the muscular tissue under loading and metabolic stress suggesting plausible systemic responses driven by the increased training volume. Similarly, the conflicting evidence in shoulder abduction, internal and external rotation could be attributed as well to the methodological diversity (i.e., no information on the testing position) and the training imbalances (i.e., exercise number imbalance) between studies (Table 2 & S2 Table in S1 File). For example, the two studies that measured shoulder abduction strength as an outcome incorporated a disproportionate number of exercises: only external rotation [35] compared to five upper limb exercises including resisted shoulder abduction [34].

The evidence regarding LL-BFRT compared to HL-RT in muscle strength of healthy individuals is conflicting, with two systematic reviews showing superiority evidence for HL-RT [6, 30] and two others not [30, 52]. Our findings for muscles proximal to the BFR application and consequently not directly under BFR, indicate very low certainty evidence that LL-BFRT and HL-RT are equally effective in improving muscle strength. Despite the methodological diversity and inconsistencies, we suggest that LL-BFRT could be used as an alternative intervention for muscle strength improvement even in muscles located proximal to the occlusion site in individuals that cannot train with higher loads.

## Effect of LL-BFRT in muscle size

Contemporary evidence suggests that LL-BFRT may elicit significant increases in muscle size when compared to LL-RT without BFR [15] and similar muscle adaptations (size) to HL-RT [6, 15, 30] in healthy adults and individuals with musculoskeletal conditions [53]. Eight weeks of rotator cuff training with LL-BFRT may induce greater increases than matched LL-RT in

the shoulder region muscle mass and in the whole upper limb where muscle mass is assessed using Dual Energy X-Ray Absorptiometry [36]. Notably, no difference between LL-RT with or without BFR was observed in pectoralis major muscle size, thickness, or girth in three studies implementing only chest press exercise [39–41]. It seems that the proximal muscle size adaptations may be plausibly driven by the total time under occlusion, a minimum volume threshold, the training period, or a systemic effect (Table 2), rather than the specificity of the exercise performed. The sensitivity of the muscle size measurement method may have played a role in the findings of our review. A significant effect of LL-BFRT was observed only in shoulder lean mass compared to LL-RT measured by DEXA in contrast to studies implementing muscle size/thickness measurements using US or tape measure. The lack of measurable between-group differences may stem from the fact that tape [54] and US [55] are not considered as reliable methods for measuring changes in regional muscle size and volume in contrast to DEXA which is considered the gold standard [56]. The measurement method may also explain the contradictory results in pectoralis major size in studies comparing LL-BFRT to HL-RT [41, 42]. Nevertheless, in line with previous research [6, 15, 30], our findings showed that LL-BFRT results in similar muscle size adaptations with HL-RT in muscles not directly under BFR conditions.

### Effect of LL-BFRT in tendon thickness

The evidence for the effect of BFRT on tendon structural and physiological adaptations is sparse in both healthy [47, 57, 58] and individuals with musculoskeletal conditions [59, 60]. In the Achilles and patellar tendons of healthy individuals, findings suggest within-group increases in tendon thickness (at 12-weeks) which were at least comparable to HL-RT without BFR [47, 57] and were consistent with our findings for the supraspinatus tendon (at 8-weeks)– despite being proximal to the BFR application. The load magnitude did not seem to play a major role in these adaptions, as the lower limb tendon studies compared LL-BFRT to HL-RT, while in the supraspinatus tendon study [35] the BFRT was compared to LL-RT. Nevertheless, given the limited number of available studies along with the evident variability in normal tendon thickness, the suggestions of loading for at least 14 weeks for optimal tendon adaptations, and the limitations of the thickness measurement methods [61–63], these results should be interpreted with caution.

### Limitations

The lack of standardized protocols, methodology, and measurement methods did not permit extended quantitative synthesis and limits the generalizability of our findings. Along with the small number of relevant studies, several other individual-study factors contributed to this limitation, including the total number of exercises performed, the application of BFR only in the upper or in both upper and lower limbs, the total time under loading and occlusion, single-limb of bilateral training, and bilateral of contralateral limb training. We also acknowledge that the limited number of the included studies did not allow performance of some of the standard analyses of a systematic review as for example the assessment of publication bias using funnel plots or related regression methods. A standardized method of BFR study reporting and protocol of application should be established in future research allowing for optimization in the evidence translation into clinical practice.

### Practical applications

Contrary to common resistance training guidelines and BFR mechanisms of action hypotheses, increases in strength and size of muscle proximal to BFR application can be achieved using

low loads. Evidence suggests that performing light LL-BFR causes systemic hypoalgesia comparable with HL-RT in healthy individuals [64] and patients with anterior knee pain [65, 66] and produces cross-over contralateral limb (upper and lower) loading adaptations [22, 34]. Plausibly, these results and the reported contralateral and whole-body cross-over effects [43] may be explained by a systemic response to LL-BFRT. Nevertheless, current assumptions are mainly based on indirect observations [43] and further research is required to evaluate these effects.

In the upper limb in muscles distal to the BFR application, the effect of LL-BFRT appears similar to HL-RT and seems to be minimally affected by variability in the intensity and the occlusive pressure implemented (Table 2). Clinicians may see this as a window of opportunity for loading exercises in populations that are not cleared or capable of using higher loads (i.e., musculoskeletal pathology). There was considerable heterogeneity in the prescribed exercise parameters which were observed to induce positive adaptations. We suggest that an important area of future research is determining the minimal exercise intensity and loading volume for beneficial proximal adaptations to occur.

## Conclusion

Low and very low certainty evidence suggests a significant increase in bench press 1RM and in shoulder flexion MVIC in favor of the LL-BFRT compared to the LL-RT without BFR, and very low certainty evidence of a significant increase in shoulder lean mass at the 8-week follow-up, but these findings should be interpreted with caution. LL-BFRT elicits comparable muscle adaptations (strength and size) in shoulder girdle muscles to both LL- and HL-RT. The minimal volume threshold and the total time under occlusion required for beneficial responses are yet to be described.

## Supporting information

**S1 File. Table 1 is the search strategy and results in databases, Table 2 is the measurement method of the outcome of interest in strength, muscle size, and tendon thickness, and Fig 1 is the Forest plots depicting studies using LL-BFRT compared to studies using LL-RT in muscle strength presenting significant statistical heterogeneity ($I^2>75\%$).**
(PDF)

**S1 Checklist. PRISMA 2020 checklist.**
(DOCX)

## Author Contributions

**Conceptualization:** Kyriakos Pavlou.

**Data curation:** Kyriakos Pavlou, Vasileios Korakakis, Christos Karagiannis, George Ploutarchou, Christos Savva.

**Formal analysis:** Kyriakos Pavlou, Vasileios Korakakis, Rod Whiteley, George Ploutarchou.

**Funding acquisition:** Rod Whiteley.

**Investigation:** Kyriakos Pavlou, Vasileios Korakakis, Christos Karagiannis.

**Methodology:** Kyriakos Pavlou, Vasileios Korakakis, Christos Karagiannis, George Ploutarchou, Christos Savva.

**Project administration:** Vasileios Korakakis, Christos Karagiannis, Christos Savva.

**Software:** Kyriakos Pavlou.

**Supervision:** Vasileios Korakakis, Rod Whiteley, Christos Karagiannis, Christos Savva.

**Validation:** Kyriakos Pavlou.

**Visualization:** Kyriakos Pavlou.

**Writing – original draft:** Kyriakos Pavlou, Vasileios Korakakis.

**Writing – review & editing:** Kyriakos Pavlou, Vasileios Korakakis, Rod Whiteley.

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
