## [Decision Letter · Decision Letter 0]

19 Dec 2022

PONE-D-22-31845The effects of upper body Blood Flow Restriction training on muscles located proximal to the applied occlusive pressure: A Systematic Review with meta-analysis.PLOS ONE

Dear Dr. Whiteley,

Thank you for submitting your manuscript to PLOS ONE. After careful consideration, we feel that it has merit but does not fully meet PLOS ONE’s publication criteria as it currently stands. Therefore, we invite you to submit a revised version of the manuscript that addresses the points raised during the review process.

ACADEMIC EDITOR: Dear Author, Please attend to all the points that need to be revised as raised by the reviewers. The decision of this manuscript is justified based on PLOS ONE’s publication criteria and not on its novelty or perceived impact.

We look forward to receiving your revised manuscript.

Kind regards,

Zulkarnain Jaafar

Academic Editor

PLOS ONE

“If accepted for publication, this research may have open access fees paid for by the Qatar National Library Open Access Research Fund. Asie from this, this research did not receive any specific grant from funding agencies in the public, commercial, or not-for-profit sectors.”

“None declared”

Reviewers' comments:

Reviewer's Responses to Questions

**Comments to the Author**

1. Is the manuscript technically sound, and do the data support the conclusions?

Reviewer #1: Partly

Reviewer #2: Partly

2. Has the statistical analysis been performed appropriately and rigorously? 

Reviewer #1: I Don't Know

Reviewer #2: No

3. Have the authors made all data underlying the findings in their manuscript fully available?

Reviewer #1: Yes

Reviewer #2: No

4. Is the manuscript presented in an intelligible fashion and written in standard English?

Reviewer #1: Yes

Reviewer #2: No

5. Review Comments to the Author

Reviewer #1: Comments to the author(s)

The authors included relevant articles after a comprehensive literature search (including manual search through the references from articles and abstracts) and provided a valuable report. Inclusion and exclusion criteria as well as number of evaluated studies are clearly stated. However, the number of included studies and the heterogeneity of the study characteristics make the results less generalizable (participant activity level [trained (7), untrained (2)], number of sessions per week [2-6 sessions per week], duration of the intervention [ 2-8 weeks], varying exercise loads [additionally in some studies the control group performed relatively higher intensity exercises (references: 35, 36, 39, 40)] as well as change in exercise load throughout the intervention program are quite heterogeneous across studies).

The paper was submitted to the journal as “Research Article”. However, the journal PLOS ONE does not consider systematic review and meta-analysis in this category. Here’s the journal statement on the subject: “Research articles must report on original research that contributes to the base of academic knowledge. Reviews, essays, opinion pieces, hypothesis papers, and other items of secondary literature are not considered.”

The manuscript presented in an intelligible fashion and written in standard English.

Abstract

-Line #18- Reformulate the sentence. For example, “Blood flow restriction combined with low load resistance training (LL-BFRT)”.

-Line #22- Change the abbreviation as “(LL-RT, HL-RT)”.

Introduction

-Line #45- It is suggested to remove the word “minimum”. The authors may type “~60-70%”.

-Line #50- The sentence may need to be reformulated. For example, “Recently, significant attention has been drawn to low load resistance training (LL-RT) combined with blood flow restriction (BFR), which involves a parallel partial restriction of the arterial flow and complete occlusion of the venous return of the exercised limb (3, 4).”

-Line #56- This paragraph is quite informative.

-Line #67- It may be more useful if the authors provide underlying reasons for these possible mechanisms.

-Line #70- The authors need to remove the word “apparent”.

-Line #73- The authors need to explain why “the large muscle groups of the upper extremity” is a possible limitation for the application of BFRT, especially compared to lower body.

Line #75- This last sentence is not a great match for the paragraph. The first two sentences in the paragraph addresses the understudied muscle groups of the upper extremity and provides possible explanations for it. The last sentence, on the other hand, is not related to these.

Line #82- Edit as “such as changes in muscle size and strength”.

Methods

-Line #93- The study objective states that “the main objective of this systematic review of randomized controlled trials (RCTs) was to evaluate in healthy individuals the effectiveness of LL-BFRT...”. However, the population for eligibility is different here in line #93. Please reformulate.

-Line #136- The authors are suggested to reformulate this sentence. Particularly, the beginning of it (“As measures of intervention effect, given that the outcomes were continuous, ...”) is not very clear.

-Line #152- The authors need to provide references for the categorization of training loads. These subgroups (> 60% 1RM for HL; >40 1RM for LL) do not agree with some of the studies referenced in the Introduction section and are relative.

-Line #154- The authors may need to explain why they are including studies with a non-exercising control group. Please also examine the study objective (at the end of the Introduction section) accordingly. This line also contradicts with the statement in line #191 (“The control group used the same exercises with the BFRT group in every study..”).

Results

-Line #181- In Table 2, third row (Brumitt et al., 2020), please indicate if the significant difference is compared to baseline levels (within-group).

-Line #181- In Table 2, 8th row (Yamanak et al., 2012), a comma (,) proceeds the p (p,0.05). Please reformulate.

-Line #181- In Table 2, 9th row (Yasuda et al., 2010) “physiological adaptations and main findings” is not clear, please reformulate.

-Line #190- Understandably the authors used the term “fatigue”, since the referenced article also used that term. However, it could be more accurate to use a term such as “volitional exhaustion” or “until participants' expressed inability to continue with the task”.

-Line #223- The referenced study (39) included 40 participants in total and had 4 groups. It could be better if the authors reformulated this sentence similar to this one: “...one study of moderate quality and high risk of bias (39) showed no difference in bench press strength between LL-BFRT (n=10) and no exercise (n=10)...”.

-Line #259- Add space between “population” and “(40)”. Please also check the whole paper for similar edits (for example line #334).

Discussion

-Line #295- The authors summarized the main findings clearly in this paragraph.

-Line #312- The authors cited the reference #44 (Centner et al., 2019), however it is not related to the statement made in that sentence. Centner et al., (2019) did not include a LL-RT group, rather they had a HL-RT group.

-Line #317- This sentence is not clear. The authors need to explain what they mean by “fatiguing effect of synergistic muscles”.

-Line #390-422- The authors stated the study limitations and conclusions clearly. The authors may include that the number of studies reviewed did not allow them to perform some of the standard analyses.

-The authors seem to have cited related and up to date references.

Reviewer #2: The manuscript aimed to investigate the effect of blood flow restriction training combined with low load resistance training (LL - BFRT) on upper limb muscle strength and the results of the meta-analysis showed that LL-BFRT had effectiveness in improving muscle strength and size in upper body muscles located proximal to the BFR stimulus in healthy adults. However, the evidence was low certain. Additionally, several issues in this manuscript also should be concerned.

Major issues：

1. The quality assessment for included papers is very important for “A Systematic Review with meta-analysis”, I suggest “Supplementary Table 2.” should not be supplementary data, it should be put in the manuscript.

2. The authors pooled the papers into a meta-analysis, but the manuscript only showed the effect size of every trial on GRAGE assessment tables. I suggest that forest flots should be used for the evaluation of the total effects and subgroups should be used for different comparisons.

3. It is hard to read Table 2. In this manuscript, I suggest the characteristics should be: Article, year; country/Region; participants characteristics, sample size (every group); intervention (for every group); duration of intervention; outcomes.

4. I suggest the Methods section should be rewritten as lots of details need to be added, ex. in line 137, “we calculated mean differences (MDs), standardized mean differences (SMDs) where more than one study was available”, it is mean only one study used (MD), more than one study used (SMD)? it’s not correct.

In lines 139-140, “forest plots were presented if aggregate pooled estimates met the sample and intervention homogeneity criteria.” There was no forest plots in this manuscript except supplementary data, besides, the detail of information on “the sample and intervention homogeneity criteria” should be mentioned.

In lines 147-150, “If considerable between-group statistical heterogeneity……… heterogeneity at baseline for participant characteristics”, we can’t direct move out one article from the meta-analysis pool just because of the high I2, the reason for high heterogeneity should be analyzed, sometimes, subgroups were suggested to be used.

5. in lines 169-170, “of which 179 articles were full text screened. Nine studies met the eligibility criteria and were included.” Two articles were included in the meta-analysis by citation searching but not of which 179 articles.

6. Supplementary table 1. only showed the search strategy in Medline, but not other databases.

Minor issues：

1. English writing should be improved.

2. In the abstract section, "Discussion" should be replaced by "Conclusion", the "LL-HT" group in the “Objective” should be replaced by "LL-RT".

3. in lines 153-154, “60% 1RM load during exercise (high-load – HL), or (ii) low intensity exercise (>40% of 1RM), or (iii) a control group (no exercise) as a comparator”, the authors should provide the reference.

4. “Author contributions” should be spilt from the “Acknowledgments” section

6. PLOS authors have the option to publish the peer review history of their article (what does this mean?). If published, this will include your full peer review and any attached files.

Reviewer #1: No

Reviewer #2: **Yes: **Xi Chen

---

## [Author Response · Author response to Decision Letter 0]

16 Feb 2023

Reviewer comments have been individually addressed, and the responses along with the changes to the text related to each response are tabluated in the uploaded "Response to Reviewers" document

---

## [Decision Letter · Decision Letter 1]

2 Mar 2023

PONE-D-22-31845R1The effects of upper body Blood Flow Restriction training on muscles located proximal to the applied occlusive pressure: A Systematic Review with meta-analysis.PLOS ONE

Dear Dr. Whiteley,

Thank you for submitting your manuscript to PLOS ONE. After careful consideration, we feel that it has merit but does not fully meet PLOS ONE’s publication criteria as it currently stands. Therefore, we invite you to submit a revised version of the manuscript that addresses the points raised during the review process.

ACADEMIC EDITOR: Dear Author, This manuscript still requires some minor changes to be made. Please attend to all the points mentioned. The decision of this manuscript is justified based on PLOS ONE’s publication criteria and not on its novelty or perceived impact.

We look forward to receiving your revised manuscript.

Kind regards,

Zulkarnain Jaafar

Academic Editor

PLOS ONE

Journal Requirements:

Reviewers' comments:

Reviewer's Responses to Questions

**Comments to the Author**

1. If the authors have adequately addressed your comments raised in a previous round of review and you feel that this manuscript is now acceptable for publication, you may indicate that here to bypass the “Comments to the Author” section, enter your conflict of interest statement in the “Confidential to Editor” section, and submit your "Accept" recommendation.

Reviewer #1: All comments have been addressed

Reviewer #2: All comments have been addressed

2. Is the manuscript technically sound, and do the data support the conclusions?

Reviewer #1: Partly

Reviewer #2: Partly

3. Has the statistical analysis been performed appropriately and rigorously? 

Reviewer #1: I Don't Know

Reviewer #2: Yes

4. Have the authors made all data underlying the findings in their manuscript fully available?

Reviewer #1: Yes

Reviewer #2: Yes

5. Is the manuscript presented in an intelligible fashion and written in standard English?

Reviewer #1: Yes

Reviewer #2: Yes

6. Review Comments to the Author

Reviewer #1: The authors have revised the manuscript sufficiently. It may be useful to take a look at the few minor comments below before consideration to publish this research.

-Line #186- One of the studies you included had 6 sessions per week. So, revise as “with 2 to 6 exercise sessions per week”.

-Line #187- The median and range values are not correct. Please revise.

-Line #310- Better strength and size adaptations compared to which training? Since this is the first paragraph of the Discussion, it is particularly important to give the reader a broad and clear main finding.

-Line #383- Revise the section “by using DEXA” or remove it completely.

-Table 2- Brumit et al., (2020) – Participants characteristics: What does “54% trained young adults” mean?

Reviewer #2: This version of the manuscript has improved so much. The "conclusion" section in the text mentions that caution should be exercised to interpret these findings due to limited evidence. I suggest that the authors mentioned in the abstract before publication.

7. PLOS authors have the option to publish the peer review history of their article (what does this mean?). If published, this will include your full peer review and any attached files.

Reviewer #1: No

Reviewer #2: **Yes: **Xi Chen

---

## [Author Response · Author response to Decision Letter 1]

5 Mar 2023

Please see the "Reply to Reviewers" document which addresses each of the reviewers' comments individually, and shows the associated changes to the text

---

## [Decision Letter · Decision Letter 2]

7 Mar 2023

The effects of upper body Blood Flow Restriction training on muscles located proximal to the applied occlusive pressure: A Systematic Review with meta-analysis.

PONE-D-22-31845R2

Dear Dr. Whiteley,

We’re pleased to inform you that your manuscript has been judged scientifically suitable for publication and will be formally accepted for publication once it meets all outstanding technical requirements.

Kind regards,

Zulkarnain Jaafar

Academic Editor

PLOS ONE

Additional Editor Comments (optional):

Reviewers' comments:

Reviewer's Responses to Questions

**Comments to the Author**

1. If the authors have adequately addressed your comments raised in a previous round of review and you feel that this manuscript is now acceptable for publication, you may indicate that here to bypass the “Comments to the Author” section, enter your conflict of interest statement in the “Confidential to Editor” section, and submit your "Accept" recommendation.

Reviewer #1: All comments have been addressed

2. Is the manuscript technically sound, and do the data support the conclusions?

Reviewer #1: Yes

3. Has the statistical analysis been performed appropriately and rigorously? 

Reviewer #1: I Don't Know

4. Have the authors made all data underlying the findings in their manuscript fully available?

Reviewer #1: Yes

5. Is the manuscript presented in an intelligible fashion and written in standard English?

Reviewer #1: Yes

6. Review Comments to the Author

Reviewer #1: Thank you authors for your contributions to the field.

This version of the manuscript is clearer.

Accept

7. PLOS authors have the option to publish the peer review history of their article (what does this mean?). If published, this will include your full peer review and any attached files.

Reviewer #1: No

---

## [Editor Report · Acceptance letter]

13 Mar 2023

PONE-D-22-31845R2 

The effects of upper body Blood Flow Restriction training on muscles located proximal to the applied occlusive pressure: A Systematic Review with meta-analysis 

Dear Dr. Whiteley:

I'm pleased to inform you that your manuscript has been deemed suitable for publication in PLOS ONE. Congratulations! Your manuscript is now with our production department. 

Kind regards, 

on behalf of

Dr. Zulkarnain Jaafar 

Academic Editor

PLOS ONE